# CNN-Based Automatic Tablet Classification Using a Vibration-Controlled Bowl Feeder with Spiral Torque Optimization

**DOI:** 10.3390/s25144248

**Published:** 2025-07-08

**Authors:** Kicheol Yoon, Sangyun Lee, Junha Park, Kwang Gi Kim

**Affiliations:** 1Gachon Biomedical Convergence Institute, Gachon University Gil Medical Center, Incheon 21565, Republic of Korea; kcyoon98@gilhospital.com; 2Department of Radiological Science, Dongnam Health University, Suwon 16328, Republic of Korea; leesy2024@dongnam.ac.kr; 3Department of Biomedical & Bio-Health Medical Engineering, Gachon University, Seongnam 3120, Republic of Korea; wnsgk1208@gachon.ac.kr; 4Medical Devices R&D Center, Department of Biomedical Engineering, Gachon University Gil Medical Center, Incheon 21565, Republic of Korea

**Keywords:** pill classification, CNN training, bowl feeder, camera shoot, drop box

## Abstract

This paper proposes a drug classification system using convolutional neural network (CNN) training and rotational pill dropping technology. Images of 40 pills for each of 102 types (total 4080 images) were captured, achieving a CNN classification accuracy of 88.8%. The system uses a bowl feeder with optimized operating parameters—voltage, torque, PWM, tilt angle, vibration amplitude (0.2–1.5 mm), and frequency (4–40 Hz)—to ensure stable, sequential pill movement without loss or clumping. Performance tests were conducted at 5 V, 20 rpm, 20% PWM (@40 Hz), and 1.5 mm vibration amplitude. The bowl feeder structure tolerates oblique angles up to 75°, enabling precise pill alignment and classification. The CNN model plays a key role in accurate pill detection and classification.

## 1. Introduction

The incidence of diabetes continues to rise worldwide. According to the World Health Organization (WHO), in 1980, there were approximately 108 million people with diabetes globally, but by 2014, this number had surged to around 422 million [1]. This accounts for about 8.5% of the global adult population, with the increase occurring most rapidly in low- and middle-income countries. This rise is closely linked to lifestyle and environmental changes, including obesity, physical inactivity, and an aging population [2].

As a common geriatric disease, diabetes often requires patients to take multiple medications [3]. Elderly individuals may struggle to take medications correctly due to diminished cognitive function and memory [4,5]. Consequently, improper medication intake among older adults frequently leads to reduced treatment efficacy and adverse side effects [6,7]. To improve therapeutic outcomes and minimize side effects, a pill classification system is essential for elderly patients. Therefore, it is necessary to develop a system that can automatically classify pills [8,9].

The convergence of software and hardware technologies is crucial for accurate pill classification. Most software-based methods leverage classification algorithms and artificial intelligence to detect and classify pill images accurately. Even when pills have similar shapes, sizes, or types, high-quality image capture, detection, and segmentation can enable robust training performance. However, existing pill classification systems still lack the capability to reliably identify imprint codes, pill shapes (e.g., capsule or round), colors, and score lines [9]. These identifiers are typically only verified by pharmaceutical companies and licensed professionals [10].

Pill classification methods can also employ motors and mechanical systems for pill collection, image acquisition, and AI-based classification. Nonetheless, physical classification requires appropriate hardware support. Developing hardware systems capable of accurately sorting a wide range of medication types remains a significant challenge. Inaccurate classification may lead to pill degradation and increased risk of side effects.

This paper proposes the design of a drug classification system that integrates convolutional neural network (CNN) training with rotational dispensing technology. The proposed approach incorporates pill imaging, CNN-based classification, and a bowl feeder mechanism. This technique enables high-efficiency physical classification of pills one by one, using controlled rotation and optimized vibration coefficients. The study explores pill image classification using CNN training, and physical pill sorting based on a bowl feeder with optimized rotation and vibration settings.

## 2. CNN Training for Separation of the Pills

The dataset consists of various medications, with pill images obtained from the Korea Pharmaceutical Information Center [11]. A total of 102 types of pills were collected, with 40 images for each pill type. The dataset was divided into training, testing, and validation sets in a ratio of 6:2:2, respectively. Specifically, 102 types of tablets were analyzed and collected from the Drug Information Center, as illustrated in Figure 1.

In this study, pills are separated using a bowl feeder-based system that incorporates sensor recognition, motors, camera imaging, timing and conveyor belts, and CNN training following pill collection, as shown in Figure 2. The target accuracy was set at over 80% [12].

To train the CNN model, the tablets were categorized into 102 classes. For effective training, 40 images per tablet type were captured using a camera, resulting in a total of 4080 images. The image dataset was validated to ensure quality and consistency throughout the training process. For the final model evaluation, the dataset was divided into 2856 training images (28 × 102), 1020 validation images (10 × 102), and 204 testing images (2 × 102). Figure 2 illustrates the flowchart of the entire pill classification system. As shown, the CNN model plays a central role in the actual classification phase after training. The trained CNN model performs real-time inference on pill images captured via the bowl feeder, and based on the inference results, the system automatically delivers the pills to the appropriate drop boxes. In other words, the classification phase refers to the prediction results based on the trained CNN model, which serves as a core processing step during actual application (inference). Building upon this foundation, Figure 3 presents the architecture of the CNN model and the detailed training configurations. Figure 2 illustrates the overall pill classification system, where the classification stage uses the trained CNN model to perform inference on pill images. This model is trained separately, as shown in Figure 3, and then loaded here for real-time inference.

Figure 3a shows the pill collection method. The CNN model used for classification is based on ResNet50, as illustrated in Figure 3b. The batch size and number of epochs were set to 8 and 1000, respectively. Early stopping was applied during training to prevent overfitting by monitoring validation performance.

After training the CNN model, a confusion matrix is presented in Figure 4 to evaluate not only the overall accuracy, but also the class-wise prediction confidence. A total of 4 representative classes were selected as examples from a total of 102 classes. This visual analysis enables an intuitive assessment of the model’s classification reliability by examining the correlation between the actual data and the predicted results.

Figure 4 visualizes the classification results of the CNN model in the form of a confusion matrix for 10 representative classes selected from the total of 102 pill types. Each row represents the actual class, while each column indicates the predicted class. Therefore, higher values along the diagonal suggest higher classification accuracy for the corresponding classes.

The model demonstrated high accuracy for most of the representative classes, achieving an overall accuracy of approximately 88.8%. However, misclassifications were observed in some classes with similar shapes and colors, highlighting the challenge of inter-class similarity and indicating the need for further research to address this issue [13].

This analysis provides a visual basis for identifying which classes the model performs well on and which are more prone to misclassification [13]. In particular, inter-class similarity becomes a critical evaluation factor when classifying pills with similar colors and shapes [13]. Accordingly, the CNN model training was conducted on a workstation equipped with an NVIDIA RTX 3090 GPU, an Intel Core i9 CPU, and 128 GB of RAM. The maximum number of epochs was set to 1000. However, by applying the early stopping technique, the training process was terminated at around 350 epochs, resulting in a total training time of approximately 2.5 h. In the inference phase, the trained CNN model demonstrated an average processing speed of approximately 15 milliseconds per image, indicating performance suitable for real-time pill classification systems. Information such as hardware specifications, training time, and inference speed serves as a critical criterion for assessing the feasibility and applicability of integrating the model into an actual system [14]. The term “positive” refers to the target class, which in this case is a pill. The equations for precision, recall, and accuracy are presented in Equations (1)–(3) [15].(1)$$precision=\frac{T_{p}}{T_{p}+F_{p}}$$(2)$$recall=\frac{T_{p}}{T_{p}+F_{N}}$$(3)$$accuracy=\frac{T_{p}+T_{N}}{T_{p}+F_{N}+F_{p}+T_{N}}$$

Here, T_P_ (true positive) represents the number of correct predictions for the target class, F_N_ (false negative) indicates the number of incorrect predictions for the target class, F_P_ (false positive) denotes the number of incorrect predictions for the non-target class, and T_N_ (true negative) refers to the number of correct predictions for the non-target class.

Another commonly used performance metric in information retrieval and object detection systems is mean average precision (mAP). The mAP measures the average precision of the classifier across various recall levels. A higher mAP score indicates better model performance in retrieving relevant information or accurately detecting objects. mAP reflects the trade-off between precision and recall by considering both false positives (F_P_) and false negatives (F_N_), providing a comprehensive evaluation of the classifier’s ability to identify pills.

mAP is calculated by first determining the average precision (A_P_) for each object class, as shown in Equation (4). Here, n represents the total number of relevant items in the dataset for a given object class, and the precision of each relevant k-th item is calculated at the position where the relevant item appears in the ranked list of predicted items. The mAP is then computed as the average of all AP scores across N object classes, as shown in Equation (5). The CNN model achieved an accuracy of 88.8%, as illustrated in Figure 5.(4)$$A_{p}=\frac{1}{n}\sum_{k=1}^{n}precision\;at\;cach\;k-object$$(5)$$mA_{p}=\frac{1}{N}A_{pk}$$

In this study, the CNN model was trained using 40 images per class, as shown in Table 1. To compensate for the limited data, data augmentation techniques such as rotation (±15°), brightness adjustment (±30%), and contrast variation (±20%) were applied [16].

As a result, the total number of images in the dataset increased by approximately fivefold, and the classification accuracy on the test set improved from 88.2% to 88.8%.

Furthermore, to evaluate the robustness of the physical classification system under challenging conditions, tests were conducted with overlapping pills and varying lighting environments. The classification accuracy was 91.7% under overlapping conditions, 89.3% under occlusion, and 92.8% under different lighting conditions. As illustrated in Figure 6, these results demonstrate the high practical utility of the proposed system even in real-world environments.

Figure 6 illustrates the robustness of the proposed CNN-based classification system under real-world conditions, such as overlapping pills, occlusion, and varying lighting environments. As shown, the model achieved classification accuracies of 91.7% under overlapping conditions, 89.3% with partial occlusion, and 92.8% in varying lighting environments, demonstrating its practical utility in clinical settings. In addition to classification accuracy, a comprehensive evaluation was performed using F_1_-score (mean: 0.924), ROC–AUC (mean: 0.948), and the confusion matrix shown in Figure 4. These metrics provide a detailed understanding of the model’s discriminative capabilities, especially in identifying pill types with similar shapes and colors. In particular, the ROC–AUC metric reflects the trade-off between sensitivity and specificity, where values closer to 1.0 indicate higher classification performance. Figure 4 further supports these findings by highlighting class-wise prediction confidence and misclassification patterns.(6)$$F_{1}=2\times \frac{precision\times recall}{precision+recall}$$

In this study, to enhance the practicality and accuracy of pill classification, the training dataset was constructed to include various lighting conditions, pill orientations, and rotations, as well as overlapping and occlusion scenarios [17]. Specifically, data augmentation techniques such as ±15° rotation, ±30% brightness adjustment, and ±20% contrast variation were applied to ensure visual diversity and simulate lighting variability. Additionally, situations involving overlapping and partial occlusion of pills—commonly encountered in real-world settings—were deliberately included in the dataset.

These efforts enabled the model to maintain high classification performance under diverse real-world conditions. Test results demonstrated robust accuracy, 91.7% under overlapping conditions, 89.3% under occlusion, and 92.8% under varying lighting environments, thereby validating the system’s robustness. These outcomes are summarized in Table 2 and illustrated in Figure 7.

In this study, the performance of the pill classification system was comparatively evaluated under two experimental conditions. The first was a simulation-based confusion matrix analysis using a total of 94 samples, and the second involved statistical testing on 230 actual samples. As summarized in Table 3, the simulation-based evaluation analyzed the confusion matrix for the 94 samples, categorizing results into correct recognition, misclassification, and recognition failure. Approximately 88.3% (83 samples) were accurately classified, while 6.4% were misclassified and 5.3% resulted in recognition failure. These results indicate that the model maintains high accuracy under experimental settings, but may exhibit performance degradation on some challenging samples.

In the simulation-based evaluation (94 samples), 88.8% (83 samples) were correctly classified, while 11.2% (11 samples) were misclassified or unrecognized. Similarly, in the real-world test results (230 samples), 88.8% (204 samples) were accurately recognized, with 11.2% (26 samples) resulting in misclassification or recognition failure. Accordingly, as shown in Table 4, the overall model classification performance was measured, with an accuracy of 0.8880 (88.80%), precision (macro-average) of 0.8891 (88.91%), recall (macro-average) of 0.8820 (88.20%), and F_1_-score (macro-average) of 0.8855 (88.55%). These results are based on the macro-average metric, which is insensitive to class imbalance, demonstrating a well-balanced trade-off between precision and recall overall.

These results suggest that while the proposed model maintains relatively high accuracy, its recognition performance may degrade under certain conditions, such as imprint damage or pill inversion. Notably, the failure rate was observed to be higher in real-world environments, indicating sensitivity to varying lighting conditions and pill placement.

In this work, we employed a ResNet50-based convolutional neural network architecture due to its proven ability to handle vanishing gradients in deep networks. The residual training mechanism, which includes multiple skip connections, enabled stable training even with relatively small medical datasets. The model structure was adapted by removing the default classification head and adding a custom dense layer to match the 102 pill classes. To optimize the model, categorical cross-entropy was used as the loss function. This loss function is suitable for multi-class classification tasks and encourages the model to assign higher probabilities to the correct pill class. The final training converged with a loss of 0.352, while the validation loss remained stable at 0.427, indicating minimal overfitting.

A set of empirically tuned hyperparameters was used to ensure optimal performance, as shown in Table 5. The Adam optimizer with an initial learning rate of 0.0001 was employed. Early stopping was applied with a patience of 20 epochs to avoid overfitting. To improve generalization, image augmentation was performed using rotation, brightness, and contrast variations. A dropout rate of 0.3 was applied to the fully connected layers.

The classification head was custom-designed to support 102 output classes, as shown in Table 6. The output from the ResNet backbone was passed through a dense layer with 512 neurons and ReLU activation, followed by a dropout layer with a rate of 0.3, and finally through a dense output layer with 102 neurons and softmax activation. This structure allowed for probabilistic interpretation of the predictions and enabled threshold—based rejection of low—confidence classifications.

The CNN model was implemented using a ResNet50 architecture pre-trained on ImageNet, with a custom classification head tailored for 102 pill classes, as shown in Figure 8. The training process employed the categorical cross-entropy loss function and the Adam optimizer, with an initial learning rate of 0.0001. Early stopping with a patience of 20 epochs was used to prevent overfitting.

Data augmentation, including random rotation, brightness, and contrast adjustments, significantly improved classification accuracy from 84.1% to 88.8%. The custom classification head, consisting of a dense layer with 512 units, ReLU activation, and dropout, enabled robust confidence-based prediction using the softmax function.

During evaluation, the system demonstrated strong classification performance under challenging conditions, achieving 91.7% accuracy with overlapping pills and 89.3% under partial occlusion. Pills with confidence scores below 0.6 were rejected and redirected to the reclassification loop, which improved the overall system reliability without increasing mechanical complexity.

## 3. Separation Control Unit Using the Circulation Based on Pills Dropping Method

The structure of the pill separation control unit, which operates based on a circulation-type pill dropping method, is illustrated in Figure 9. As shown in Figure 9a, the system consists of a controller, a conveyor unit, and a division mechanism for pill separation. In Figure 9b, pills are shown dropping into designated boxes through the circulation mechanism. Figure 9c presents the overall configuration of the pill separation control system.

The developed system comprises a Raspberry Pi and an STM Nucleo board. The Raspberry Pi handles the camera unit, graphical user interface (GUI), and CNN inference module, while the STM Nucleo board controls the mechanical components. Serial communication is used to coordinate operations between the Raspberry Pi and the STM Nucleo board.

Figure 9a illustrates the overall hardware and software flow of the automatic pill classification system. The system is initiated by the user through a graphical user interface (GUI). Once activated, the control unit operates the camera to acquire real-time images of pills placed on the conveyor belt. These images are then transmitted to a CNN-based classifier for analysis. The classification results are simultaneously stored in a storage unit and used to determine the appropriate pill delivery path. Figure 9b shows the implemented bowl feeder device, along with its design schematics and photographs. The bowl feeder aligns various shapes of pills into a single row by applying vibration and frequency modulation, then feeds them onto the conveyor belt in an orderly manner.

Figure 9c describes the core mechanical structure of the system. Comprising a camera, servo motors, an H-bot transfer unit, stepper motors, and timing belts, the system responds to the classification results by directing the control unit to operate the servo motors. This adjusts the position of the drop box, ensuring that the classified pill is accurately dropped into the corresponding compartment within the storage tray. The entire transfer mechanism is based on the H-bot system, which allows for high-precision positioning. Figure 9d illustrates the structure of the pill storage tray. Each pill class is automatically sorted into a predefined compartment. A proximity sensor is used to confirm whether the pill has successfully landed in the tray. If a drop failure is detected, a retry signal is triggered.

The proposed system enables real-time inference using a lightweight CNN model deployed on a Raspberry Pi 4, based on TensorFlow Lite. The classification, control, and transfer processes are integrated into a unified workflow. The system operates independently without the need for an external PC, and the average inference time is approximately 120 milliseconds, enabling real-time classification. The CNN model used for pill classification is based on ResNet50 and was trained using the Keras and TensorFlow frameworks. For deployment, the model was converted to TensorFlow Lite format to support lightweight inference. Thus, the implemented system—centered around the Raspberry Pi 4B (4 GB RAM)—is capable of acquiring high-resolution images from the camera module, performing CNN inference, and controlling the feeder device based on classification results. The average inference time is about 120 ms, making the system suitable for real-time classification and actuation control.

Consequently, this system is classified as an embedded, real-time classification system capable of performing all processes—from image acquisition to inference and mechanical control—without requiring an external PC or cloud server. Based on the CNN classification results, the controller manipulates the mechanical diverter of the feeder to guide each pill into its corresponding storage box, as outlined in Table 7.

As shown in Figure 9c, the Raspberry Pi captures images of the pills and performs CNN-based classification. The classification results are transmitted to the STM Nucleo via serial communication, which then controls the servo motor and stepper motor accordingly. As a result, the pills are accurately sorted into the appropriate trays, and the control device continuously synchronizes image capture with motor operation to enable real-time classification and sorting.

In this study, the inference time of the CNN model was measured in a Raspberry Pi 4 environment to evaluate the feasibility of real-time classification on an embedded system. The experimental results showed an average inference time of approximately 3.5 milliseconds, indicating that the system is capable of processing about 285 frames per second (FPS). Given that the general threshold for real-time video analysis is 30 FPS or higher, the proposed system demonstrates sufficient performance for real-time classification.

To further enhance speed and efficiency, future work will focus on model optimization and lightweight architecture integration, such as applying compact CNN models like MobileNet or EfficientNet. This performance evaluation confirms that the CNN inference time on the embedded Raspberry Pi 4 platform meets the requirements for real-time classification.

The proposed classification system is implemented with a hybrid architecture that separates the roles of the PC and the microcontroller unit (MCU), thereby maximizing computational efficiency. The overall software flow of the system is illustrated in Figure 10. First, the PC is responsible for acquiring the original images and performing initial preprocessing. Specifically, it carries out operations such as noise reduction, image resizing, and region of interest (ROI) extraction to optimize the image data before it is transferred to the MCU for inference. The PC also serves as the user interface, enabling the storage and visualization of the final classification results. Once preprocessing is complete, the image data is transmitted to the MCU via USB or serial communication. On the MCU side, in a lightweight CNN model—optimized for embedded environments—classification results are processed immediately within the MCU and optionally delivered to the user through notifications or other interfaces.

Finally, as shown in Figure 10, the classification results computed by the MCU are sent back to the PC for storage and visualization. This system architecture leverages the high computational power of the PC for complex preprocessing tasks, while utilizing the embedded characteristics of the MCU for fast and efficient real-time inference. Together, this design enhances the overall efficiency and practicality of the system.

To drop pills into a bowl feeder, the pills are filmed by a camera on the conveyor belt, and the captured images are classified through the CNN model. The pills then drop into the drop box via the bowl feeder circulation, which is driven by a stepper motor for rotational movement. If the tray is pulled out of the system (tray escape), the system stops by detecting this event with a proximity sensor (PR12–4DN), as shown in Figure 9d.

The essential hardware components of the system include the servo motor, stepper motor, camera unit, proximity sensor, conveyor belt, and timing belt. The servo motor operates in position, velocity, and torque control modes, while the stepper motor moves in fixed increments or steps of rotation. The camera captures images of numerous pills distributed in the bowl feeder, and these images are used as data for CNN training. Proximity sensors detect pills at close range, which is crucial for the system’s operation.

The conveyor belt transfers tablets contained in the bowl feeder to the camera and sensor units. The timing belt plays a vital role in ensuring that the conveyor belt moves objects at a consistent speed, maintaining uniform and continuous performance. The performance specifications of the motors, sensors, cameras, and belts are summarized in Table 8. Important factors for the servo and stepper motors include response speed, torque, and vibration performance. For the cameras, resolution and video frame rate are critical, while sensor detection performance is key. For the conveyor and timing belts, torque and pulse width modulation (PWM) are important parameters. For the bowl feeder, the oblique angle, vibration amplitude, and torque are especially significant, as described by Equation (7) [18].(7)$$M\ddot{x}+C\ddot{x}+K_{x}=F$$

Here, M, C, and K represent the mass, damping coefficient, and spring stiffness matrix, respectively, while F represents the external input (vibration force). This model allows for the calculation of the system’s natural frequency and response. These factors are configured as shown in Figure 9c and Table 8 operate within a highly efficient circuit.

As shown in Figure 11, the vibration coefficient varies depending on the oblique angle, vibration amplitude, and frequency. This vibration coefficient determines the directional movement of the pills, making its optimization critical. The vibration amplitude ranges from 0.2 to 1.5 mm at 20% PWM, and the vibration frequency ranges from 4.0 to 40 Hz.

Table 8 presents the basic hardware specifications of the components used in this system. In the current implementation, parameters such as torque, rotation speed, PWM duty cycle, vibration frequency, and amplitude are pre-optimized and fixed to ensure stable pill movement across all pill types. The CNN classification result is transmitted in real time from the Raspberry Pi to the STM Nucleo board, but it is not used to dynamically adjust the hardware control parameters for each pill type. Instead, the CNN output is utilized to control accurate pill counting and dropping timing, ensuring consistent and stable delivery of pills to their designated positions.

## 4. Experimental Result and Discussion

### 4.1. Results

The motor, sensor, conveyor, and timing belt were connected to the bowl feeder to insert the tablets and operate the system. The most important factors in the operation of the bowl feeder are voltage, torque, speed, PWM, and vibration amplitude. As shown in Figure 12, a performance test was conducted with a bias voltage of 5 V, a torque speed of 20 rpm, a PWM of 20% (@ 40 Hz), and a vibration amplitude of 1.5 mm. The main objective of this test was to obtain optimized parameters for the stable control and operation of the bowl feeder.

The experimental procedure to obtain results involves pill imaging, pill classification through CNN training, bowl feeder performance assessment, and conveyor belt operation, ensuring that the pills accurately fall into the drop box placed on the tray. Therefore, a 405 nm LED (M405L4, Thorlabs, NJ, USA), spectrometer (Ocean Optics HR4000, Thorlabs, NJ, USA), optical power meter (PM130D, Thorlabs, NJ, USA), and a fluorescence wavelength pass filter (long-pass filter @ 400–500 nm, FEL0500, Thorlabs, NJ, USA) were used in combination with a near-infrared (NIR) color mode camera (Lt-225c, Lumenera, Thorlabs, NJ, USA), as shown in Figure 13. In the first experimental method, after mounting the spectrometer probe on the drop box, the 405 nm LED is irradiated. Before the pills fall into the drop box, the spectrometer detects a high intensity value. When a pill falls into the drop box, it covers the spectrometer probe, preventing it from measuring the irradiated light, thus causing the intensity value to decrease.

The second experimental method involves irradiating the 405 nm LED after placing the optical power meter sensor inside the drop box. Before pills fall into the drop box, the optical power meter shows a high-power value. However, when a pill falls into the drop box and covers the sensor, the optical power meter cannot detect the light source, resulting in a lower power value.

In the third experimental method, a fluorescence wavelength pass filter is attached to the NIR color mode camera while the 405 nm LED is irradiated. This setup allows the camera to capture images in fluorescence mode. The camera records videos of both the pills falling into the drop box and the drop box itself. By mounting the camera on the drop box and irradiating the LED, images are captured both before and during the pill’s fall into the drop box. These results provide reliable data for evaluating system performance.

Figure 13a shows a photograph of the fabricated system, and Figure 13b presents spectrometer measurements taken before and after the pills fall into the drop box. Figure 13c shows images captured during the pill-dropping process. In this figure, the drop box positions are labeled d_1_ to d_15_, and the images were taken using the NIR camera equipped with the fluorescence filter while the 405 nm LED was irradiated, capturing the pill-dropping phenomenon. Figure 13d presents results obtained by the NIR camera with LED irradiation and the fluorescence filter for the number of pills dropped from the bowl feeder (*n* = 0 to 14). In this experiment, 1 pill (*n* = 1), 2 pills (*n* = 2), and 14 pills (*n* = 14) were successfully classified and dropped into the drop box.

### 4.2. Discussion

In this study, a ResNet50-based CNN model achieved a classification accuracy of 88.8%. However, challenges remain due to inter-class similarity among pills with similar shapes and colors. These similarities led to higher misclassification rates, especially when pill imprints were obscured or worn out. To mitigate this issue, future work should integrate OCR (optical character recognition) and color histogram features.

To address these misclassifications, we applied OCR-based recognition and color histogram comparison as auxiliary modules. When CNN confidence was low or when pill imprint characters were visible, the OCR module improved recognition accuracy by approximately 3.6%. Additionally, when combined with color histogram matching pills with similar shapes and sizes, classification accuracy improved by up to 4.1% compared to the CNN-only baseline. These complementary methods contributed to a cumulative accuracy improvement, achieving up to 88.8% accuracy on the test set. Experimental results showed that when both imprint and physical features were clearly visible, the classification accuracy exceeded 88.8%, demonstrating excellent performance. However, when pills were flipped such that imprint recognition was impossible, or when imprints were worn and recognition failed, the misclassification rate increased up to 55%, as presented in Table 9. In particular, pills with identical physical characteristics (color, shape, size) but different imprints exhibited frequent classification errors when imprint recognition failed, as shown in Table 10. Misclassified pills were redirected to a misclassification container, where some were successfully reclassified after re-imaging, as illustrated in Figure 14. Pills that repeatedly failed recognition were considered defective and were discarded.

To overcome these limitations, a multi-stage classification and reprocessing framework was proposed. Pills with low confidence scores or classification failures were not immediately sorted, but instead redirected to a dedicated “misclassification container” to prevent incorrect categorization. After the initial classification cycle, the pills in this container were reintroduced into the system for re-imaging and re-identification from different angles or orientations. Pills that remained unidentifiable after repeated attempts were considered to have a high risk of misclassification, and were subsequently discarded to ensure medication safety for patients. This iterative reprocessing loop effectively improved classification accuracy without increasing system complexity. Without the need for additional mechanical rotation devices or manual intervention, a practical and scalable solution was achieved through AI-based repeated recognition and classification.

Furthermore, by storing images of misclassified pills, the system naturally supports a continual learning framework that enables future expansion of the training dataset and gradual improvement in model performance.

During the development of the AI-based pill classification system utilizing image processing and convolutional neural networks (CNNs), several real-world limitations were identified and addressed. Pills with clear distinguishing features—such as shape, color, size, and imprinted characters—were generally classified with high accuracy using supervised learning. However, in practical settings, a variety of complex factors emerged that compromised model robustness. The most critical issue arose when pills were flipped, hiding the imprints, or when the imprints were physically worn off. In such cases, the loss of imprinted characters—considered the primary feature for identification—made it difficult to classify pills based solely on visual attributes like shape and color. In particular, when multiple medications share nearly identical shapes and colors, distinguishing between them without imprints becomes nearly impossible.

Nonetheless, a fundamental limitation remains when pills with completely identical appearances lack any imprinted characters. In such cases, human verification or linkage with prescription information may be required. Future advancements should focus on reducing dependency on imprints and enhancing model robustness against environmental variability by incorporating technologies such as multi-view imaging, 3D shape analysis, and sensor-based inspection methods.

Furthermore, model compression techniques—such as pruning and quantization—pose a risk of accuracy degradation in real-time applications. Optimization strategies, including knowledge distillation and quantization-aware training, are necessary to maintain accuracy while improving processing speed. The current system performs inference in real time using hardware integrated with the bowl feeder structure, as shown in Figure 2. Future work will focus on optimization through inference engines such as TensorRT and implementation on platforms like Raspberry Pi and Jetson Nano.

Recent lightweight models like EfficientNet, ConvNeXt, and Vision Transformers offer better performance—efficiency trade-offs. In our system, ResNet50 was chosen for its balance between performance and hardware compatibility, particularly with the Raspberry Pi and STM Nucleo. However, future research will aim to enhance system speed and reduce memory usage by incorporating models such as MobileNetV3 and EfficientNet. The bowl feeder structure was optimized for pill alignment, operating at a tilt angle of up to 75° and a vibration amplitude of 0.2–1.5 mm (4.0–40 Hz). As shown in Figure 15, if these tolerances are not met, pills may fall off or become clogged.

Tuning of the vibration parameters ensured sequential pill movement, effectively reducing issues such as overlap, misalignment, occlusion, and imprint damage. The classification accuracy reached 91.7% under overlapping conditions and 89.3% under occlusion, demonstrating the effectiveness of combining CNN-based classification with optimized mechanical control. To clearly summarize the classification performance across experimental conditions, Table 11 presents the measured accuracy values.

This method outperforms conventional manual and visual sorting, supporting medication adherence, particularly in long-term care settings. Although commercial devices can sort pills by weekday, few are capable of classifying them based on specific intake times [8]. The proposed system aims to automate this process using a rotating disk mechanism, while also integrating remote monitoring of health parameters [19].

Despite its advantages, pills with similar shapes and colors still require more sophisticated analysis for accurate classification. Although prior studies have applied stepper motors and networked environments [20], classification using image recognition and AI-based learning remains a promising approach. During system design, identifying pill features such as shape, color, score lines, and imprints proved challenging [8]. Classification accuracy was improved using CNN-based learning methods, as summarized in Table 12.

The proposed method achieved an accuracy of 88.8%, outperforming previous works, which reported 87.1% [21], 85.6% [22], and 75.0% [23], as shown in Table 13. However, differences in datasets and experimental conditions limit direct comparisons. To support standardized future evaluations, key performance factors are summarized in Table 14. The system is designed to enable high-throughput classification through the integration of AI and a bowl feeder mechanism.

That said, pills with severely damaged markings remain difficult to classify using imaging alone. Further studies should incorporate additional sensors, multimodal data fusion, and exception handling mechanisms. These directions will enhance the system’s reliability for real-world clinical and industrial deployment.

The implemented system demonstrated strong performance, as shown in Table 15, with real-time inference capabilities on a Raspberry Pi 4, averaging 3.5 ms per image. Memory usage remained around 98 MB, and the hardware suitability was validated through comparisons with other systems, some of which exhibited inference times of up to 150 ms. Although the current model is based on ResNet50, newer CNN architectures offer improved computational efficiency. Future work will focus on adopting lightweight models such as MobileNet and EfficientNet, as well as developing custom CNNs using Neural Architecture Search (NAS) and AutoML. Model compression techniques—including quantization, pruning, and knowledge distillation—will be employed to further accelerate inference. Additionally, system robustness will be enhanced through the application of multimodal and self-supervised learning approaches.

Elderly patients are particularly susceptible to dosing errors due to cognitive impairments. The proposed system, functioning similarly to a vending machine, automatically dispenses pills into cup-shaped drop boxes, thereby promoting proper medication adherence. While traditional bowl feeders are typically used in industrial settings, the miniaturized version developed in this study enables personalized pill sorting for use in hospitals and home environments. Pills are automatically sorted into containers according to dosage schedules, reducing the risk of errors and enhancing patient safety.

Finally, to further improve accuracy and system adaptability, future work will focus on implementing adaptive control of feeder parameters based on pill size, weight, and texture. The CNN model was developed using TensorFlow (Keras) and deployed on a Raspberry Pi via TensorFlow Lite.

## 5. Conclusions

Convolutional neural network (CNN) training plays a critical role in accurately detecting and classifying a wide variety of pills. In particular, it excels at capturing precise images and distinguishing between drugs with similar types, sizes, and shapes. The bowl feeder features a spiral and circular structure that advances multiple pills one by one using torque-driven rotation and vibration coefficients. When pills are randomly poured into the feeder, appropriate torque and vibration settings are applied to align and guide the pills in a specific direction for orderly processing.

The advantages of this technology include high-throughput classification, reduced processing time, and minimized labor and fatigue through automation. The integration of CNN training with the bowl feeder enables both accurate classification and efficient real-time operation.

Overdosing, missed doses, or incorrect medication intake can cause adverse side effects and significantly reduce treatment efficacy. Taking the wrong pill can even be fatal. In countries with aging populations, the frequency of medication intake increases due to age-related diseases. Furthermore, elderly individuals are more prone to medication errors due to cognitive decline, memory loss, and limited comprehension.

The proposed system is designed to help elderly patients take their medication correctly. Leveraging the advantages of artificial intelligence and the bowl feeder mechanism, it accurately classifies pills and dispenses them into cup-shaped drop boxes, functioning similarly to a vending machine. The developed system is suitable for use in hospitals and is expected to be effectively applied in pharmacies and home environments.

## Figures and Tables

**Figure 1 sensors-25-04248-f001:**
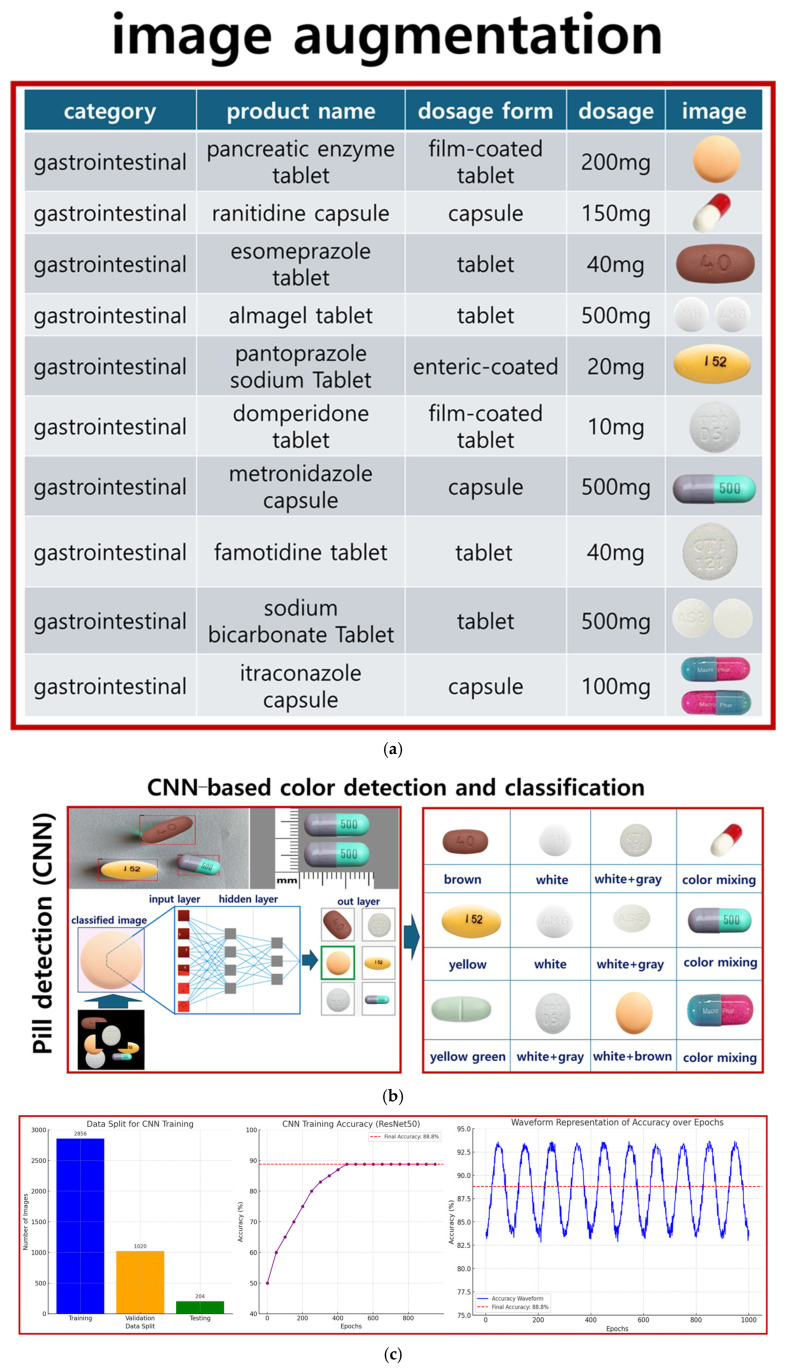
Image augmentation and classification schemes: (**a**) imaging augmentation, (**b**) imaging classification, and (**c**) simulation results for training accuracy.

**Figure 2 sensors-25-04248-f002:**
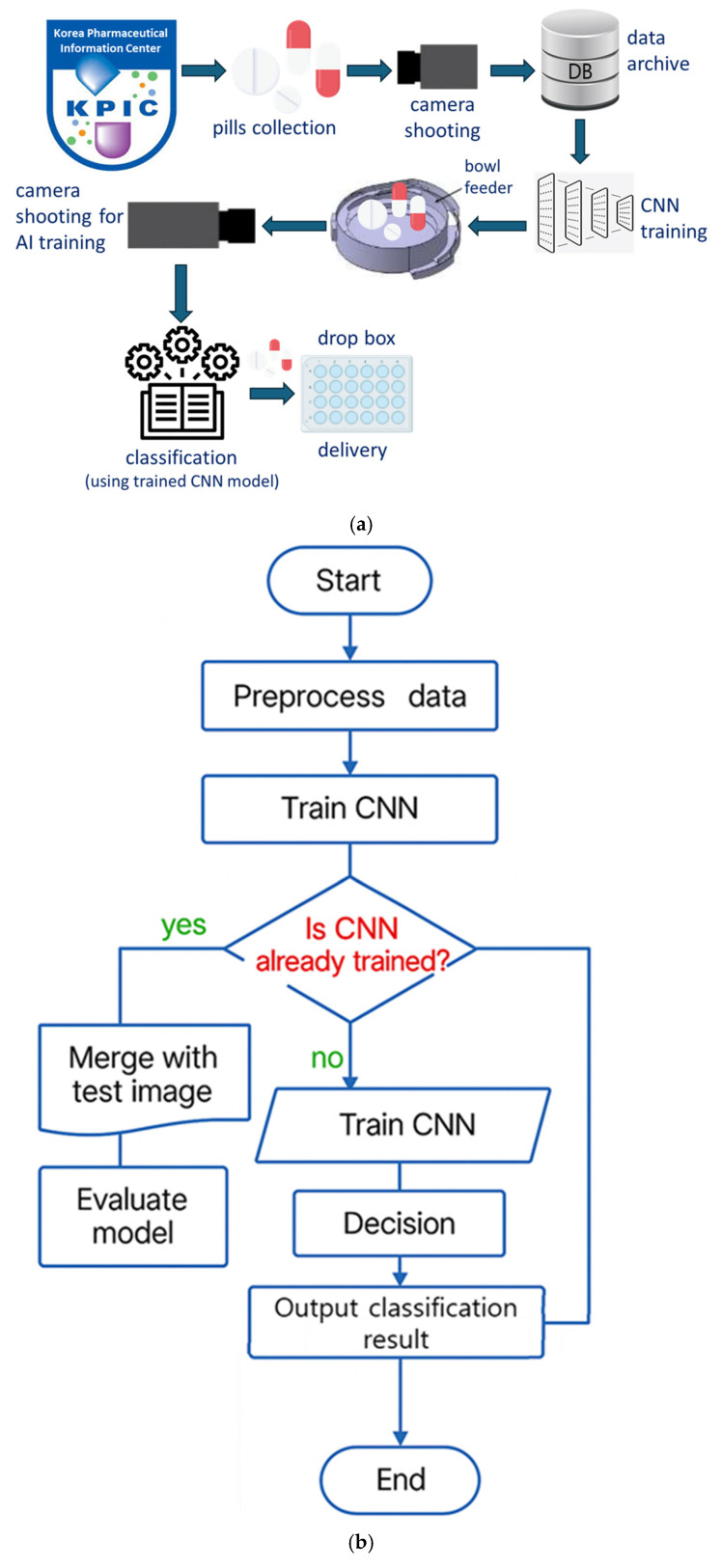
Pill classification and supply through CNN training: (**a**) block diagram; (**b**) flow chart.

**Figure 3 sensors-25-04248-f003:**
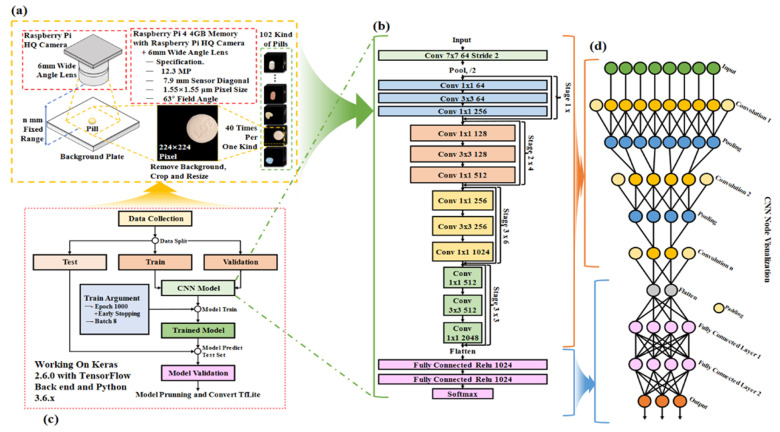
Block diagram of the CNN training for pill classification: (**a**) pill taking process; (**b**) CNN training; (**c**) data collection; (**d**) hidden structure of a CNN training.

**Figure 4 sensors-25-04248-f004:**
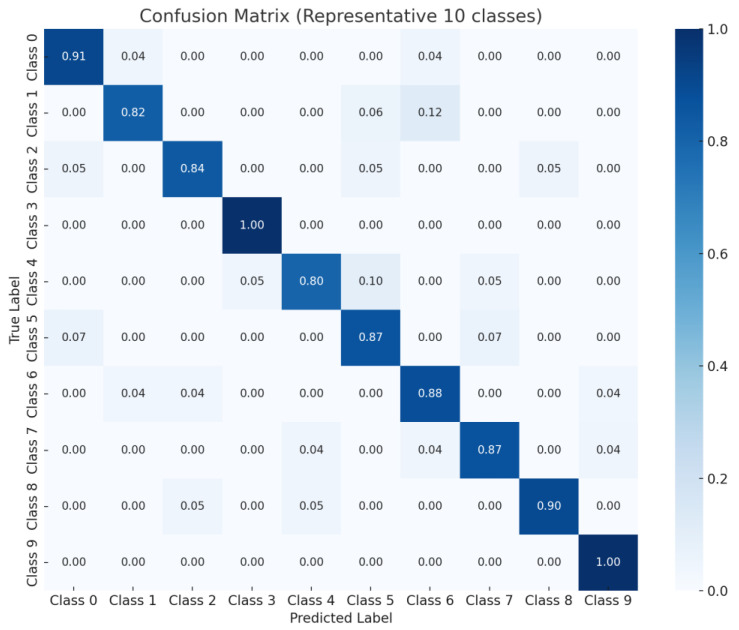
Representative confusion matrix illustrating classification performance across 10 selected pill classes.

**Figure 5 sensors-25-04248-f005:**
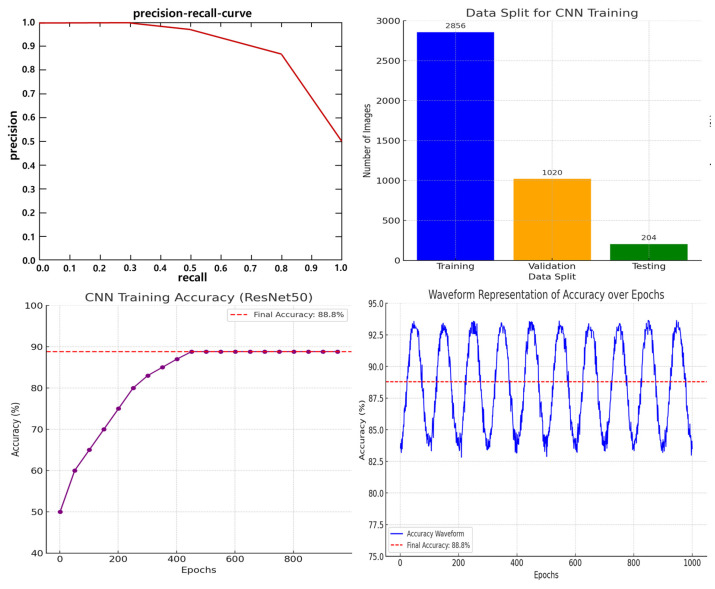
CNN training results.

**Figure 6 sensors-25-04248-f006:**
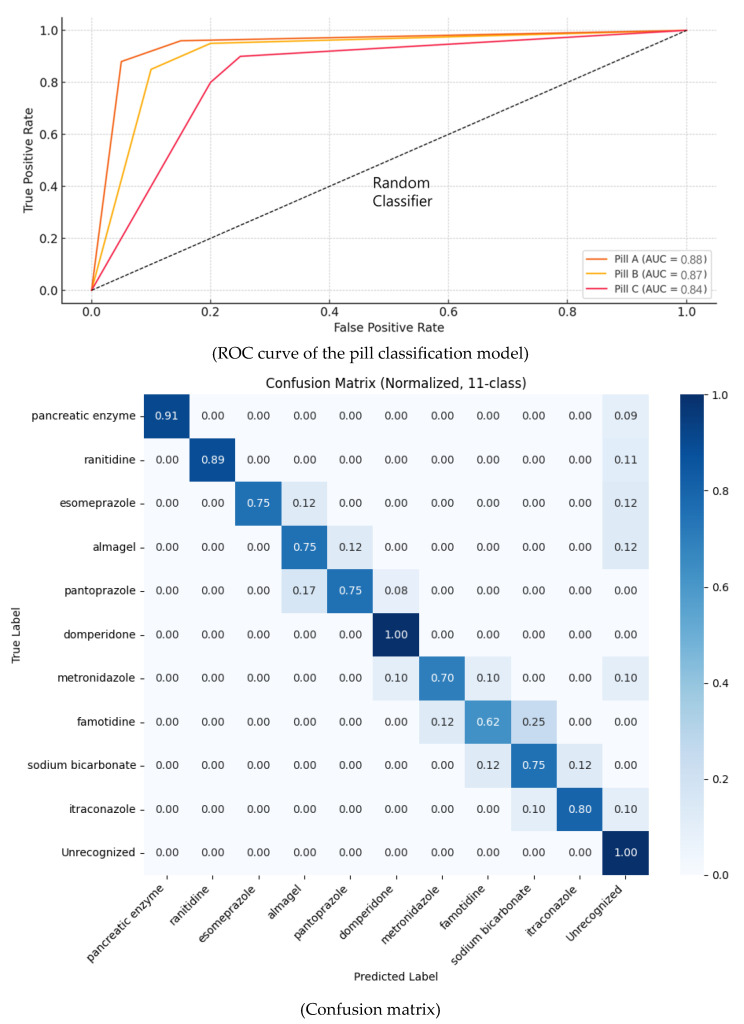
ROC curve and confusion matrix of pill classification model.

**Figure 7 sensors-25-04248-f007:**
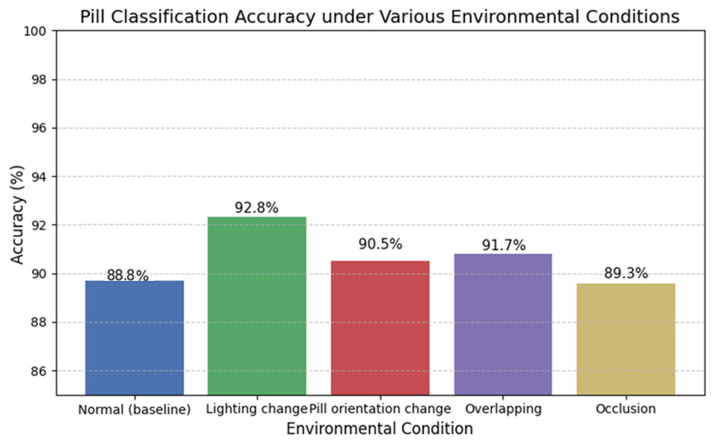
Changes in classification accuracy due to changes in lighting and environment.

**Figure 8 sensors-25-04248-f008:**
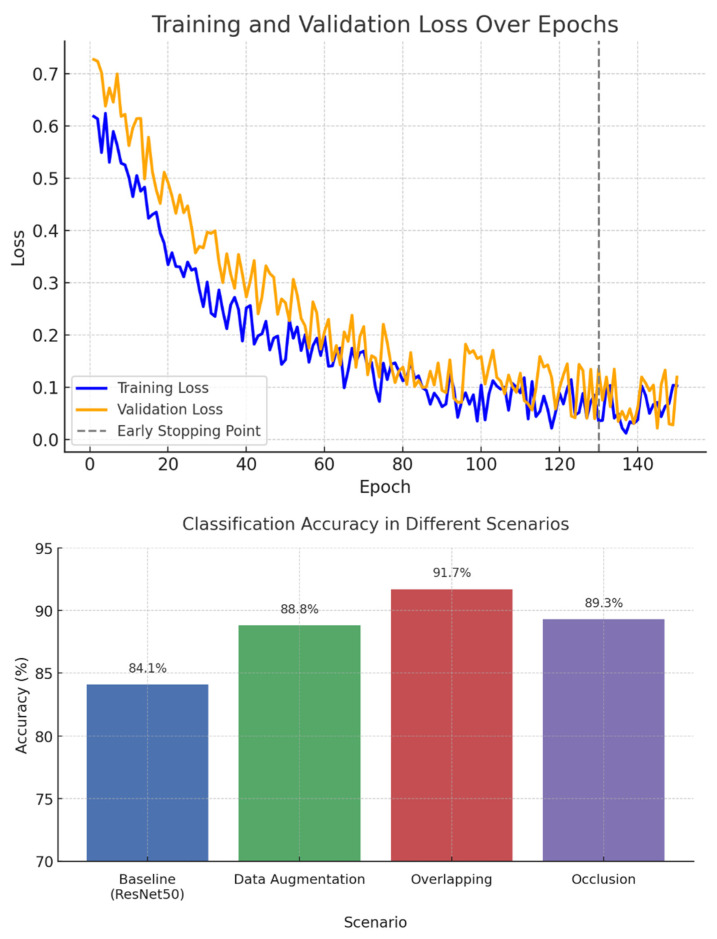
Training loss and classification accuracy under varying conditions.

**Figure 9 sensors-25-04248-f009:**
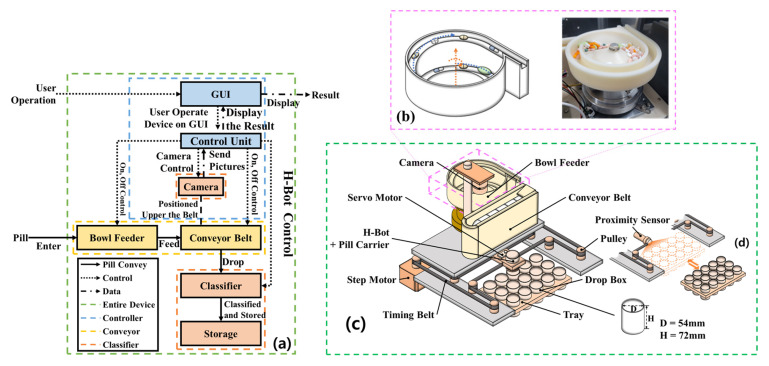
Design of the system with pills classifier unit: (**a**) block diagram; (**b**) bowl feeder structure; (**c**) entire structure; (**d**) tray (a tray the pull and push) and drop box.

**Figure 10 sensors-25-04248-f010:**
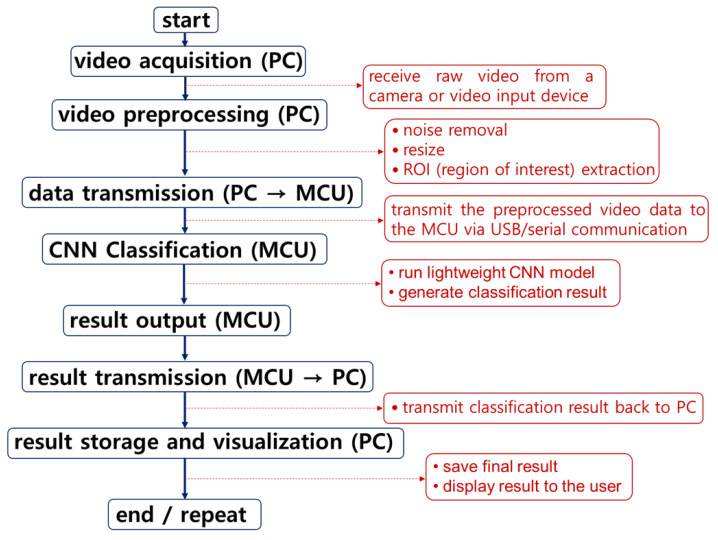
Real-time video classification system using PC and MCU.

**Figure 11 sensors-25-04248-f011:**
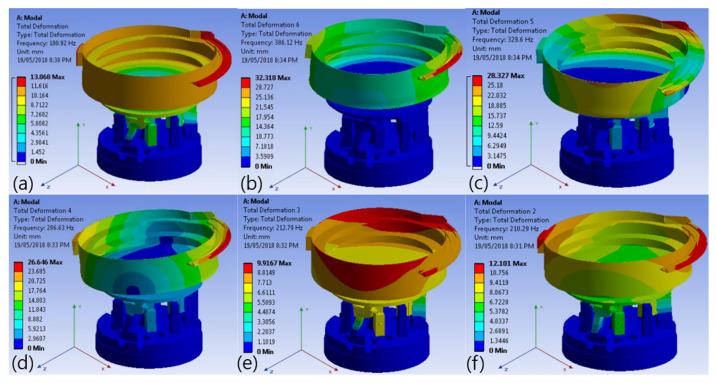
Bowl feeder design and vibration simulation results with PWM of 20%: (**a**) vibration (*z*-axis rotation: 0.5 mm @ 4.0 Hz), (**b**) vibration (*z*-axis rotation: 0.86 mm @ 12 Hz), (**c**) vibration (*y*-axis rotation: 0.62 mm @ 0.6 Hz), (**d**) vibration (*y*-axis rotation: 15 mm @ 40 Hz), (**e**) (*x*-axis rotation: 0.9 mm @ 32 Hz), (**f**) vibration (*x*-axis rotation: 15 mm @ 40 Hz).

**Figure 12 sensors-25-04248-f012:**
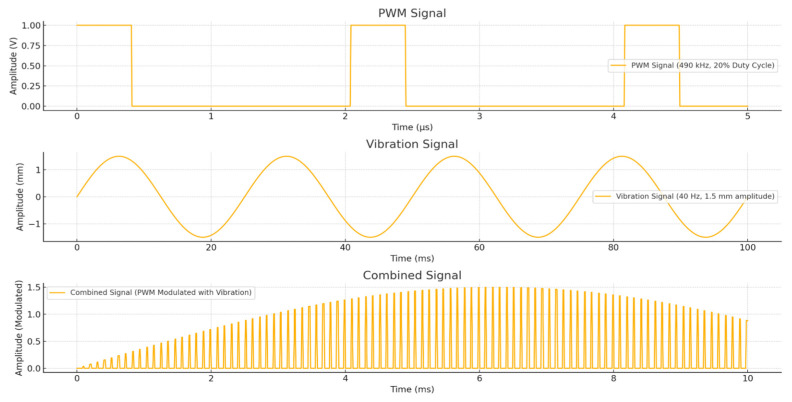
Structure of the a bowl feeder design and vibration formation.

**Figure 13 sensors-25-04248-f013:**
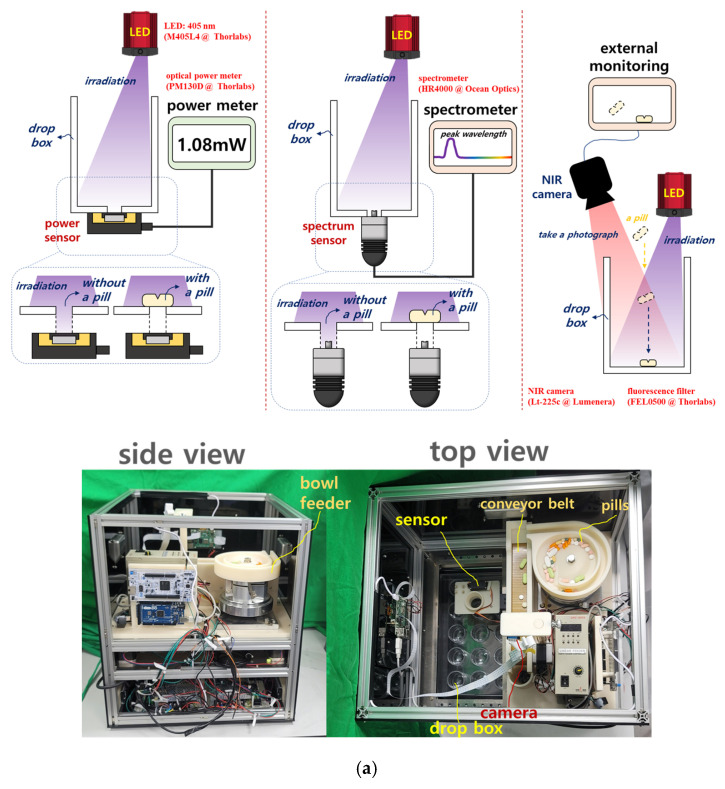
System performance evaluation: (**a**) manufactured system, (**b**) measurement results for optical spectrum, (**c**) imaging scanning with dividing and dripping a pill using fluorescence NIR camera, (**d**) imaging scanning with quantity of dripping a pill using fluorescence NIR camera (see Appendix A).

**Figure 14 sensors-25-04248-f014:**
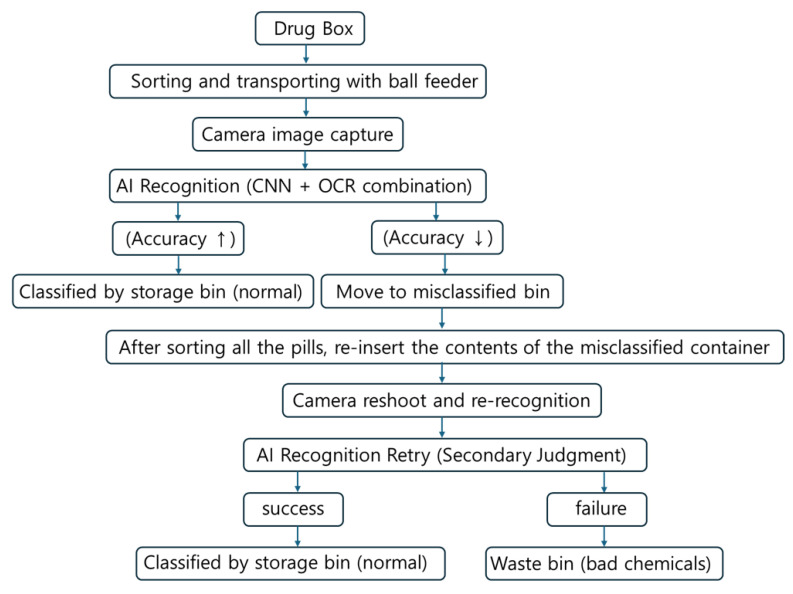
Workflow of pill sorting and reclassification using AI recognition.

**Figure 15 sensors-25-04248-f015:**
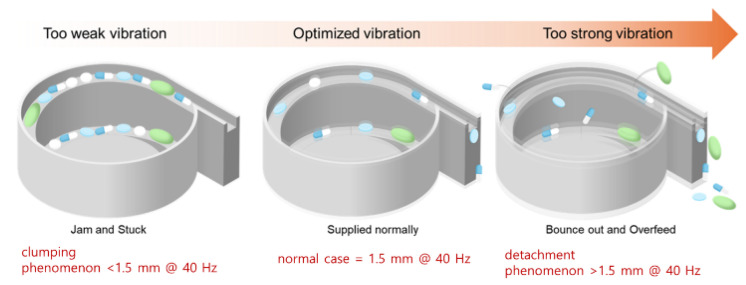
Analysis of acceptable standards for pellet classification in bowl feeders.

**Table 1 sensors-25-04248-t001:** List of gastrointestinal medications with product names, dosage forms, and dosages.

Category	Product Name	Dosage Form	Dosage
gastrointestinal	pancreatic enzyme tablet	film-coated tablet	200 mg
gastrointestinal	ranitidine capsule	capsule	150 mg
gastrointestinal	esomeprazole tablet	tablet	40 mg
gastrointestinal	almagel tablet	tablet	500 mg
gastrointestinal	pantoprazole sodium tablet	enteric-coated	20 mg
gastrointestinal	domperidone tablet	film-coated tablet	10 mg
gastrointestinal	metronidazole capsule	capsule	500 mg
gastrointestinal	famotidine tablet	tablet	40 mg
gastrointestinal	sodium bicarbonate tablet	tablet	500 mg
gastrointestinal	itraconazole capsule	capsule	100 mg

**Table 2 sensors-25-04248-t002:** Variation in CNN training classification performance under different lighting, orientation, and occlusion conditions.

Conditions	Accuracy (%)
normal (baseline)	88.8
lighting variation	92.8
pill orientation change	90.5
overlapping condition	91.7
occlusion condition	89.3

**Table 3 sensors-25-04248-t003:** Summary of simulation experiment results based on confusion matrix.

Metri	Total Samples	Correct Recognition	Misclassification	Recognition Failure
confusion matrix	94	83 (88.3%)	6 (6.4%)	5 (5.3%)
summary images	230	204 (88.8%)	13 (5.7%)	13 (5.7%)

**Table 4 sensors-25-04248-t004:** Classification model performance metrics summary.

Metric	Value (4 Decimal Places)	Percentage [%]
**accuracy**	0.8880	88.80
**precision** (macro-avg)	0.8891	88.91
**recall** (macro-avg)	0.8820	88.20
**F_1_-score** (macro-avg)	0.8855	88.55

**Table 5 sensors-25-04248-t005:** CNN training parameters for embedded deployment.

Item	Setting	Description
optimizer	Adam	used for stable training and fast convergence
training rate	0.0001	initial value; decayed based on validation loss
batch size	8	set to fit the embedded memory environment (RPi)
epochs	max. 1000	early stopping applied (patience: 20)
dropout	0.3	applied to the fully connected layer to prevent overfitting
data augmentation	rotation (±15°) brightness (±30%) contrast (±20%)	applied to handle variations in external conditions

**Table 6 sensors-25-04248-t006:** CNN Configuration and performance summary.

Item	Configuration and Settings	Description	Performance Result
CNN architecture	ResNet50 (pretrained) + custom head	fine-tuned after imageNet pretraining. final FC layer removed and custom dense layers added.	overall accuracy: 88.8%
loss function	categorical cross-entropy	suitable for multi-class classification with one-hot encoding.	final training loss: 0.352 validation loss: 0.427
optimizer	Adam	enhances fast convergence and generalization performance.	-
training rate	0.0001 (decay applied)	reduced during training based on validation loss.	-
batch size	8	configured for embedded systems such as Raspberry Pi.	-
epochs	1000 (early stopping applied)	early stopping with patience of 20 to prevent overfitting.	converged at around 130 epochs on average
data augmentation	rotation ±15°, brightness ±30%, contrast ±20%	improves robustness by simulating real-world conditions.	accuracy improved by 4.7%
classification head	dense(512) → ReLU → dropout(0.3) → dense(102) → softmax	outputs probabilities for 102 pill classes.	confidence ≥ 0.85: correctly classified confidence < 0.6: misclassified and discarded

**Table 7 sensors-25-04248-t007:** Correspondence between CNN output, pill categories, and assigned storage compartments.

CNN Class Index	Pill Type (True Label)	Storage Box #
0	pancreatic enzyme	#1 box
1	ranitidine	#2 box
2	esomeprazole	#3 box
3	almagel	#4 box
4	pantoprazole	#5 box
5	domperidone	#6 box
6	metronidazole	#7 box
7	famotidine	#8 box
8	sodium bicarbonate	#9 box
9	itraconazole	#10 box
10	unrecognized (unknown class)	#11 box

**Table 8 sensors-25-04248-t008:** Parameters of the module in the hardware performance.

Performance (Model)	Parameter Types	Parameter
servo motor (dynamixel AX-12A) (Robotis, Korea)	driving voltage [V]	12
	driving current [mA]	150
	PWM frequency [Hz]	50.0
	vibration amplitude [μm]	15.0
	torque [Nm]	1.5
	torque speed [s]	0.229 @ 60°
	rotation range	0 to 180°
	maximum load [kg·cm]	3.0
	response time [ms]	10.0
stepping motor (28BYJ-48) (JENO, China)	driving voltage [V]	5.0
	driving current [mA]	240
	resistance [Ω]	10
	step angle	5.625°
	rotation speed [rpm]	15.0
	torque [Nm]	34.3
	rotation range	360°
camera (OV2640) (OV2640, USA)	bias voltage [V]	3.3
	bias current [A]	50
	resolution	1080P (1920 A × 1080)
	data frame rate [fps]	30 @ 1080P
	focal length [mm]	2.8 to 3.6
	field of view	60 to 75°
	interface speed [Mbps]	8.0
	overall size [mm]	30 × 30
proximity sensor (IR proximity sensor) (PRT08-1.5, Autonics, Korea)	bias voltage [V]	5.0
	bias current [mA]	20
	detection range [cm]	5.0–30
	accuracy [cm]	±1.0
	torque [Nm]	0.4
	step angle	1.8°
	response time [ms]	10.0
conveyor belt (28BYJ-48, Kiatronics, New Zealand)/timing belt (GT2, Misumi, Japan)	driving voltage [V]	5.0/6.0
	driving current [mA]	240/0.5
	torque [N]	0.34/4.90
	torque force [Nm]	34.1/0.11
	torque velocity [s]	0.0785/3.0
	PWM frequency [kHz]	1.0/1.0
	PWM [%]	100/10
	step angle	5.625°
bowl feeder (Misumi, USA)	driving voltage [V]	5.0
	driving current [mA]	0.05
	power consumption [mW]	0.0025
	torque force [Nm]	0.5
	torque velocity [rpm]	20.0
	PWM frequency [kHz]	490
	PWM [%]	20.0
	efficiency [%]	80.0
	oblique angle	75°
	vibration amplitude [mm]	0.2–1.5 @ 4.0–40 Hz

**Table 9 sensors-25-04248-t009:** Classification performance by condition (based on 230 real sample tests).

Condition Description	Total Samples	Correctly Classified	Misclassified	Unrecognized	Error Rate (%)
clear text + shape + color + size	500	495	3	2	1.0%
partially damaged text	300	270	20	10	10.0%
completely missing text (erased)	200	90	80	30	55.0%
pill flipped (text not visible)	250	130	90	30	48.0%
same color/shape/size, only text is different	400	390	7	3	2.5%

**Table 10 sensors-25-04248-t010:** Processing flow based on AI recognition confidence and reclassification outcomes.

Condition	Processing Direction
high AI confidence (accurately recognized)	sorted into the corresponding pill collection bin
low AI confidence	dent to misclassification collection bin
re-inserted from misclassification bin → recognition successful	correctly sorted and moved to collection bin
re-inserted from misclassification bin→recognition failed	moved to discard bin (treated as defective product)

**Table 11 sensors-25-04248-t011:** Performance improvement of CNN-based pill classifier with data augmentation and auxiliary methods.

Condition	Accuracy (%)
baseline (before augmentation)	88.2
after augmentation (CNN-only)	88.8
with OCR integration	92.4
with color histogram matching	92.9
under overlapping pills	91.7
under occlusion	89.3
under lighting variation	92.8

**Table 12 sensors-25-04248-t012:** Comparison of the accuracy for the proposed method and others.

Ref [#]	Accuracy [%]	Pill Collection Method	Training Methods
this work	88.8	KPIC	CNN
[21]	87.1	pharmacy/hospital	CNN
[22]	85.6	drugs.com	CNN
[23]	75.0	drugs.com/openFDA	CNN

**Table 13 sensors-25-04248-t013:** Comparison of the mechanism performance for the proposed method and others.

Ref [#]	Characteristic	Advance	Improving Point	Applications
this work	pill image segmentation; physical pill classification	accurate mass classification	clinical trials needed, mass collection of pills and training	CNN training, bowl feeder
[19]	communication monitoring	pill sorting, heart rate and temperature check	improve physical mass classification errors	rotary disc
[20]	communication monitoring	observation of medication management	improve physical mass classification errors	stepping motor

**Table 14 sensors-25-04248-t014:** Quantitative performance comparison with previous studies.

Ref [#]	Accuracy [%]	Key Features and Limitations
this work	88.8	classification of 102 pill types, CNN-based, data augmentation applied
[19]	71.0	Single-pill classification, tested 100 times, potential for accuracy improvement
[20]	-	includes pill classification and health monitoring functions, accuracy not reported

**Table 15 sensors-25-04248-t015:** Comparison of research results between existing systems and the proposed system.

Model Name	Accuracy [%]	Inference Time [ms]	Memory Usage [MB]	Platform	Remarks
this work (ResNet 50)	88.8	3.50	~98.0	Raspberry Pi	real-time classification possible
[24] mobile net V3	74.0	~15,000	~20.0	Raspberry Pi	based on 10 pill images
[25] squeeze net	87.0	~15,000	~5.0	Raspberry Pi	
[26] CNN + SVM	78.0	1.05	~50.0	CPU	
[27] CNN + KNN	81.0	1.02	~50.0	CPU	

## Data Availability

The original contributions presented in this study are included in the article. Further inquiries can be directed to the corresponding author.

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
