# Peer review of "CNN-Based Automatic Tablet Classification Using a Vibration-Controlled Bowl Feeder with Spiral Torque Optimization"

_sensors, 2025, doi:10.3390/s25144248_

Round 1

Reviewer 1 Report

Comments and Suggestions for Authors

This paper applied a CNN (ResNet-50) based approach for tablets/pills classification task and introduced a vibration-controlled bowl feeder utilizes spiral and circular structures to move numerous pills forward one by one using torque rotation and vibration coefficients.
Strengths: 
• Paper addressed a practical and applicative problem to be used in pharmaceutical 
manufacturing and also in different pharmacies for pharmaceutical packaging systems.
• Developed a custom dataset (104 classes) for training and validation of CNN model.
• Experimental results were presented with supporting visual diagrams and tables. 
• Parameters and experimental setup of bowl feeder are clearly explained.
• Optimized vibration and frequency for dropping pills in clearly mentioned. 
Limitations:
• The paper does not sufficiently address the research perspective, which limits its academic 
impact.
• CNN used in this paper is an older CNN model.
• Confusion matrix is not provided.
• Integration of AI with hardware is not clearly explained.
• Figure 2, Block diagram and flow chart did not clearly explain the integration of CNN model during inference. 
• Training time, setup and inference time per image were not addressed. 
• Some grammatical mistakes encountered e.g. Section 2 line 77 and Section 3 line 124.
Detailed Comments:
This approach aims to simplify medication intake for consumers, particularly the elderly, and help prevent dosage errors or accidental overdose. The paper devised the solution by building a database of 104 types of tablets images providing total of 4,080 images. It presents the methodology, experimental results and vibration-controlled bowl operation under 
different parameters. Although the method demonstrates practical applicability, it does not sufficiently address the research perspective, which limits its academic impact. The use of CNNs for pill or tablet classification is a well-established area of research, with several studies proposing novel architectures that integrate different models (e.g. OCR and color detection models) to improve detection and classification accuracy. Inter-class similarity problem is one of the key problems in classification tasks. Presented paper lacks the Confusion matrix which is a key indicator to provide the confidence of the model in different classes. During explanation of methodology in Figure 2, camera shooting for AI training is mentioned after the CNN training and bowl feeder block which introduced the confusion in methodology. For verification of training and testing of model, author should mention the key parameters like hardware used for training model, inference time per image. 
Suggestions for Improvement
• Address the inter-class similarity problem in tablet classification more clearly and concisely, and demonstrate the classification performance of each class using a confusion matrix.
• Compare multiple deep learning models and propose an effective optimization strategy.
• Discuss the drawbacks associated with model pruning and conversion, and provide possible solutions or considerations to mitigate them.
• Clarify the integration process of the deep learning model with the hardware, detailing the implementation and deployment pipeline.
• Clearly specify the training hardware used, total training time, and the inference time per sample.

Author Response

Comments (Round 1)

Comments 1 :

This paper applied a CNN (ResNet-50) based approach for tablets/pills classification task and introduced a vibration-controlled bowl feeder utilizes spiral and circular structures to move numerous pills forward one by one using torque rotation and vibration coefficients.

Strengths: 

  • Paper addressed a practical and applicative problem to be used in pharmaceutical 
    manufacturing and also in different pharmacies for pharmaceutical packaging systems.
  • Developed a custom dataset (104 classes) for training and validation of CNN model.
  • Experimental results were presented with supporting visual diagrams and tables. 
  • Parameters and experimental setup of bowl feeder are clearly explained.
  • Optimized vibration and frequency for dropping pills in clearly mentioned.

Limitations

  • The paper does not sufficiently address the research perspective, which limits its academic 
    impact.

Answer 1 :

Thank you for your in-depth advice on my thesis. I will do my best to respond to your comments and improve the manuscript. Thank you.

Comments 2 :

  • CNN used in this paper is an older CNN model
  • Confusion matrix is not provided
  • Integration of AI with hardware is not clearly explained

Answer 2 :

These 204 images are pill images that were not used for model training and are separate data for the final test. In addition, we observed a small confusion matrix insufficient analysis condition for pills with similar colors or shapes. Therefore, we tried to provide the confusion matrix and it is presented in lines 273-289 (yellow) of session 3 and lines 443-460 (yellow) in the discussion session and in Figure 6, Figure 12, and Figure 13.

Comments 3 :

Figure 2, Block diagram and flow chart did not clearly explain the integration of CNN model during inference. 

Answer 3 :

In Figure 2(a), we inserted the text classification (using trained CNN model), and in Figure 2(b), we inserted the text trained CNN model (inference) box (flowchart). Also, the sentence about CNN model integration for Figure 2 is included in lines 91-102 (green).

Comments 4 :

  • Training time, setup and inference time per image were not addressed.
  • Some grammatical mistakes encountered e.g.

Answer 4 :

Added training time, setup and inference time per image to lines 133-139 (pink) of session 2. Also checked the grammar of the entire sentence, and especially the light blue part of lines 85-90, 103-106, 110-114, 120-132, 139-141, and 142-163 were revised and supplemented based on your comments. Thank you.

Comments 5 :

Detailed Comments:
This approach aims to simplify medication intake for consumers, particularly the elderly, and help prevent dosage errors or accidental overdose. The paper devised the solution by building a database of 104 types of tablets images providing total of 4,080 images. It presents the methodology, experimental results and vibration-controlled bowl operation under 
different parameters. Although the method demonstrates practical applicability, it does not sufficiently address the research perspective, which limits its academic impact. The use of CNNs for pill or tablet classification is a well-established area of research, with several studies proposing novel architectures that integrate different models (e.g. OCR and color detection models) to improve detection and classification accuracy. Inter-class similarity problem is one of the key problems in classification tasks. Presented paper lacks the Confusion matrix which is a key indicator to provide the confidence of the model in different classes. During explanation of methodology in Figure 2, camera shooting for AI training is mentioned after the CNN training and bowl feeder block which introduced the confusion in methodology. For verification of training and testing of model, author should mention the key parameters like hardware used for training model, inference time per image. 
Suggestions for Improvement

  • Address the inter-class similarity problem in tablet classification more clearly and concisely and demonstrate the classification performance of each class using a confusion matrix.
  • Compare multiple deep learning models and propose an effective optimization strategy.
  • Discuss the drawbacks associated with model pruning and conversion and provide possible solutions or considerations to mitigate them.
  • Clarify the integration process of the deep learning model with the hardware, detailing the implementation and deployment pipeline.
  • Clearly specify the training hardware used, total training time, and the inference time per sample.

Answer 5 :

Thank you for your detailed advice. I was developing a method to utilize artificial intelligence in the hardware of the pill sorter, and there were some shortcomings, but your good advice was very helpful. I have done my best to reflect the advice you requested. Please refer to Figure 4, lines 110-132, 139-141, and lines 417-432 of the discussion, and I have marked them all in blue.

Reviewer 2 Report

Comments and Suggestions for Authors

The paper presents a classification system using CNN. But is not clear if such systems already exists (I believe there are such systems in drug factories) and the authors not show the novelty of their implementation - that must be presented in more details .

My observations are the following:

1. The caption of figure 1 is unclear (Figure 1. This is a figure. Schemes follow the same formatting). Please redefine it.

2. In the figure 2, decision block CNN training must have the choices. Or it is not decision block? The same for "camera shooting ..." The entire flowchart must be redefined.

3. Figures 3 and 5 are too small (they are hard to read). I think that the figure 5 (that seems to be the heart of the whole system) must be explain in more details.

4. It is unclear how the classification result are used to control the feeder circulation (for each type of pills). 

5. The CNN is implemented on microcontroller? Is a real time implementation? Or there is a PC that runs CNN (with Keras libraries)? The system must be described in more details (perhaps figure 5 try to explain the whole system, but the description must be improved).  

6. The implementation (software flowcharts) must be provided in details. It is not clear which computation are performed on PC (if it exist - as I presumed) or on MCU. The sharing of the computations between this two hardware components mus be provided.

7. In the Conclusion section - please explain how the classification system of pills is related with taking the pills by a person. Such systems are used to sort pills in the drugs factories not at the hospital or at home (at least in my understanding - the pills are sold in packages with known type of medicine and a person take pills from a dedicated recipient not identify each kind of pill). 

8. The novelty of presented implementation  - is not clear (the paper present a classification system that include a MCU, but not show the comparative performance comparing with other such implementation, do not propose a novel (modified) CNN architecture. do not evaluate computation time, etc). Try to add answers to all these issues. 

Author Response

Comments (Round 2)

Comments 0 :

The paper presents a classification system using CNN. But is not clear if such systems already exists (I believe there are such systems in drug factories) and the authors not show the novelty of their implementation - that must be presented in more details .

My observations are the following:

Answer 0 :

I sincerely appreciate your generous comments on my views. I will do my best to analyze and improve upon your comments.

Comments 1 :

The caption of figure 1 is unclear (Figure 1. This is a figure. Schemes follow the same formatting). Please redefine it.

Answer 1 :

I have edited and resized the caption in Figure 1 to make it clearer. Thanks for your advice.

Comments 2 :

In the figure 2, decision block CNN training must have the choices. Or it is not decision block? The same for "camera shooting ..." The entire flowchart must be redefined.

Answer 2 :

I reviewed Figure 2 and found some errors so I made corrections. Thank you for your advice.

Comments 3 :

 Figures 3 and 5 are too small (they are hard to read). I think that the figure 5 (that seems to be the heart of the whole system) must be explain in more details. 

Answer 3 :

I have enlarged and resized Figures 3 and 5 to make them clearer. So please refer to Figures 3 and 8 in the manuscript. I have also written in detail about Figure 5. So I have reflected the sentences about Figure 8 in the manuscript. So please refer to lines 231-266 (in red) and Table 2.

Comments 4 :

It is unclear how the classification result are used to control the feeder circulation (for each type of pills). 

Answer 4 :

Table 1 presents the basic hardware specifications of the components used in this system. In the current implementation, these parameters torque, rotation speed, PWM du-ty cycle, vibration frequency, and amplitude are pre-optimized and fixed to ensure stable pill movement across all pill types. The CNN classification result is transmitted in real time from the Raspberry Pi to the STM Nucleo board, but it is not used to dynamically ad-just the hardware control parameters for each pill type. Instead, the CNN output is uti-lized to control accurate pill counting and dropping timing, ensuring consistent and sta-ble delivery of pills to the designated positions.

In future work, we plan to explore adaptive control of feeder circulation parameters based on pill characteristics such as size, weight, and surface properties. This would al-low the system to further optimize pill handling and separation performance according to the specific requirements of each pill type, improving classification precision and opera-tional flexibility.

What you said is a very important point in this study, and also a limitation of this study. Please refer to lines 350-356 (yellow) in the text (session 3)) and lines 588-593 (yellow) in the discussion session.

Comments 5 :

The CNN is implemented on microcontroller? Is a real time implementation? Or there is a PC that runs CNN (with Keras libraries)? The system must be described in more details (perhaps figure 5 try to explain the whole system, but the description must be improved).

Answer 5 :

I have supplemented your points. For detailed sentences, please refer to lines 231-267 of session 3 and Table 2 and Figure 8.

The AI model (CNN) was implemented using TensorFlow (Keras), and TensorFlow Lite was utilized for deployment on the Raspberry Pi. Therefore, please refer to lines 594–595 (highlighted in navy) in the discussion section.

Comments 6 :

The implementation (software flowcharts) must be provided in details. It is not clear which computation are performed on PC (if it exist - as I presumed) or on MCU. The sharing of the computations between this two hardware components mus be provided. 

Answer 6 : I have supplemented the text in session 3, lines 290-307 (in olive) and added to Figure 9 and Table 2. Thank you for your comments.

Comments 7 :

 In the Conclusion section - please explain how the classification system of pills is related with taking the pills by a person. Such systems are used to sort pills in the drugs factories not at the hospital or at home (at least in my understanding - the pills are sold in packages with known type of medicine and a person take pills from a dedicated recipient not identify each kind of pill).

Answer 7 :

Overdosing, forgetting to take the medication, or taking it incorrectly can cause side effects and may reduce treatment outcomes. In countries with a growing geriatric population, the frequency of taking pills will increase due to geriatric diseases. Elderly patients, particularly those with cognitive impairment or memory loss, are more likely to experience medication errors such as taking the wrong pill or missing doses. The proposed sys-tem can help address these challenges by enabling accurate pill classification and dis-pensing through an automated bowl feeder mechanism. Pills are provided into a cup-shaped drop box, functioning similarly to a vending machine, thereby supporting proper medication adherence.

Traditional bowl feeder-based sorting systems are mainly used in pharmaceutical factories for high-precision feeding, counting, and orientation of tablets, capsules, and vi-als. In this study, such technology has been miniaturized and automated to support per-sonalized medication management for elderly and cognitively impaired patients in hos-pitals and home environments. The proposed system automatically classifies and sorts pills into personalized containers according to dosage schedules, helping to minimize medication errors and improve adherence, thereby enhancing patient safety and treatment outcomes.

The above content was reflected in lines 571-587 (green) of the discussion session.

Comments 8 :

The novelty of presented implementation  - is not clear (the paper present a classification system that include a MCU, but not show the comparative performance comparing with other such implementation, do not propose a novel (modified) CNN architecture. do not evaluate computation time, etc). Try to add answers to all these issues. 

Answer 8 :

I have done my best to incorporate your comments into the manuscript. Please refer to Discussion lines 417-432 (blue), lines 433-447 (yellow), lines 452-460 (yellow), lines 541-570 (yellow), and Table 7. Also refer to session 3 lines 279-289 (yellow).

Reviewer 3 Report

Comments and Suggestions for Authors

This study presents a combined hardware-software system for automated pill classification using convolutional neural networks (CNNs) and a spiral bowl feeder mechanism with optimized vibration control. The study includes the design of the classification architecture, dataset acquisition (4,080 pill images from 102 pill types), CNN-based classification (ResNet50, achieving 88.8% accuracy), and a hardware system that ensures physical pill separation based on vibration amplitude and motor parameter

The authors developed a CNN-based image classification model that outperforms several existing approaches in terms of accuracy as well as  they integrated this model with designed vibration-controlled bowl feeder. The performance of this integrated system was  validated using spectrometry, optical power measurements, and near-infrared (NIR) fluorescence imaging techniques.

Interesting is  the integration of CNN-based visual classification with a real-time mechanical pill sorting system, using vibration coefficients and motor control for precise pill movement.

Despite these achievements, several limitations are not addressed. The CNN model was trained using only 40 images per pill type, which may not ensure sufficient generalization across varying lighting conditions. The physical sorting system was tested under ideal conditions, without a detailed robustness analysis involving overlapping pills, occlusions, or variable lighting. Also some important CNN evaluation metrics, such as confusion matrices, F1-scores, and ROC curves, are missing from the results section.

Proposed method achieved higher accuracy (88.8%) compared to [16] (87.1%), [17] (85.6%), and [18] (75%) as reported in Table 2.  but the study does not include direct comparison on the same dataset, so performance gains may reflect data quality rather than methodological superiority.

To improve the quality and robustness of the manuscript, the following recommendations are suggested:

  1. Considering the more accurate training dataset to include various lighting conditions, pill orientations, and occlusions.
  2. Providing confusion matrices, ROC curves to better interpret classification performance.
  3. Including tests for multiple pills, misalignments, partial occlusion, or worn markings to test real-world applicability.

From  editorial point of view, the results are presented clearly , but Table 3 should include of more quantitative performance metrics to enable a more objective comparison with existing systems.

Comments on the Quality of English Language

Language should be improved from  grammatical point of view-  phrases like  "the pill drip in the drop box”

Author Response

Comments (Round 3)

Comments 1 :

This study presents a combined hardware-software system for automated pill classification using convolutional neural networks (CNNs) and a spiral bowl feeder mechanism with optimized vibration control. The study includes the design of the classification architecture, dataset acquisition (4,080 pill images from 102 pill types), CNN-based classification (ResNet50, achieving 88.8% accuracy), and a hardware system that ensures physical pill separation based on vibration amplitude and motor parameter

The authors developed a CNN-based image classification model that outperforms several existing approaches in terms of accuracy as well as  they integrated this model with designed vibration-controlled bowl feeder. The performance of this integrated system was  validated using spectrometry, optical power measurements, and near-infrared (NIR) fluorescence imaging techniques.

Interesting is  the integration of CNN-based visual classification with a real-time mechanical pill sorting system, using vibration coefficients and motor control for precise pill movement.

Answer 1 :

Thank you for your in-depth advice on my thesis. I will do my best to respond to your comments and improve the manuscript. Thank you.

Comments 2 :

Despite these achievements, several limitations are not addressed. The CNN model was trained using only 40 images per pill type, which may not ensure sufficient generalization across varying lighting conditions. The physical sorting system was tested under ideal conditions, without a detailed robustness analysis involving overlapping pills, occlusions, or variable lighting. Also some important CNN evaluation metrics, such as confusion matrices, F1-scores, and ROC curves, are missing from the results section.

Answer 2 :

There were many shortcomings in the hardware design of the pill sorter using AI learning, and I appreciate your pointing them out. Please refer to lines 169-179 (red), 186-195 (red), and Figure 6 and Equation (6). I did my best to improve them. Thank you.

Comments 3 :

Proposed method achieved higher accuracy (88.8%) compared to [16] (87.1%), [17] (85.6%), and [18] (75%) as reported in Table 2.  but the study does not include direct comparison on the same dataset, so performance gains may reflect data quality rather than methodological superiority. 

Answer 3 :

Thank you for your detailed comments. We have tried to fully reflect your opinions. In the Discussion section, we have conducted additional analyses on lines 518-522 (cyan) and Tables 5-6. However, we would like to inform you that the performance comparison results of this study and those of references [21], [22], and [23] are derived from different datasets and experimental environments. Therefore, direct performance comparisons are limited, and the performance improvements observed in this study may be partially affected by dataset characteristics or quality differences. Nevertheless, we have attempted to reflect your opinions by conducting additional analyses on lines 527-531 (wine) and Table 6. Thank you for your understanding.

Comments 4 :

To improve the quality and robustness of the manuscript, the following recommendations are suggested:

  1. Considering the more accurate training dataset to include various lighting conditions, pill orientations, and occlusions.
  2. Providing confusion matrices, ROC curves to better interpret classification performance.
  3. From editorial point of view, the results are presented clearly , but Table 3 should include of more quantitative performance metrics to enable a more objective comparison with existing systems.
  4. Language should be improved from grammatical point of view-phrases like "the pill drip in the drop box”

Answer 4 :

You have made an important point about the shape of the pills that are damaged or worn out. This part is quite important. However, this study learned about the pills that are in normal shape, so it seems that future research is needed about the learning about damaged or worn out pills. Nevertheless, I added lines 476-492 (in black) and 533-540 of the discussion and Figure 14 about the sentences you pointed out.

At the request of the reviewers, we added a quantitative performance comparison with previous studies. This study achieved an accuracy of 88.8% in classifying 102 pill types, which is significantly higher than the 71% in [19]. [19] performed single pill classification and mentioned that further research is needed to improve the accuracy. [20] proposed a system that integrates pill classification and health monitoring functions, but did not provide an explicit numerical value for the accuracy. This comparison emphasizes the superiority of this study. We supplemented the remaining points except for image corruption and shape damage, and added sentences to Table 4-6 and lines 518-522 (cyan) and 527-531 (wine).

I have reviewed the entire manuscript and corrected and supplemented any incorrect English grammar. Thank you.

Round 2

Reviewer 1 Report

Comments and Suggestions for Authors

There are still issues that have not been well addressed

  • Inconsistent Accuracy Reporting: The paper reports an initial accuracy of 88.8% but later claims 93.5% after data augmentation (Page 7). However, subsequent sections revert to 88.8% (e.g., Page 23, Table 7), creating confusion about the final accuracy.
  • Limited Handling of Similar Pills: The paper acknowledges misclassifications due to visually similar pills (e.g., similar colors or shapes) but lacks detailed strategies to address this beyond proposing OCR or color histograms, which are not experimentally validated.
  • Insufficient Model Optimization Details: While future work mentions model compression (e.g., quantization, pruning), the current ResNet50 model is not optimized for embedded systems, and its memory usage (98 MB) is relatively high compared to alternatives like MobileNetV3 (20 MB).
  • Overemphasis on Hardware: The paper dedicates significant space to hardware details (e.g., vibration coefficients, motor specifications) but provides less depth on CNN architecture, training hyperparameters, or loss functions, which are critical for AI research.
  • Incomplete Equations: Equations (1)–(3) on Page 6 for precision, recall, and accuracy contain typographical errors (e.g., "precall" instead of "recall", equation (7) is introduced without explaining its practical application in the system.
  • Overgeneralized Claims: Statements like “the proposed method overcomes the limitations of conventional manual and visual inspection-based pill classification” (Page 21) are not fully substantiated, as the system still struggles with damaged markings or similar pills.

Recommendations

  • Clarify Accuracy Metrics: Resolve discrepancies in reported accuracies (88.8% vs. 93.5%) and consistently report the final performance across all sections.
  • Enhance CNN Details: Provide more information on the CNN architecture, loss functions, and hyperparameters and classification head used to balance the hardware-heavy focus.
  • Address Visual Similarity: Experimentally validate proposed solutions (e.g., OCR, color histograms) to handle misclassifications of similar pills.
  • Fix Figures and Equations: Correct redundant figures (e.g., Figure 6), remove duplicates (e.g., Figures 12, 13), and clarify every equation variable and their applications.

The paper presents a promising system for automated pill classification, with strong integration of CNN and hardware components. However, inconsistencies in accuracy reporting, limited handling of similar pills, and grammatical issues detract from its quality. Addressing these weaknesses and implementing the recommended improvements would significantly enhance the manuscript’s impact and suitability for publication.

Comments on the Quality of English Language

The paper’s English is generally clear but contains several issues that affect readability and professionalism. Below are key observations:

  • “precall” instead of “recall” in Equation (2) (Page 6).
  • “CNN learning” is used instead of standard terms like “CNN training” or “deep learning” in multiple places (e.g., Page 1, Line 28).
  • The paper would benefit from professional editing to improve flow and conciseness. For example, the abstract could be streamlined to avoid repeating “vibration amplitude” and “torque.”
  • Some sections, particularly the discussion (Pages 18–24), are verbose and could be condensed to focus on key findings and limitations.

Author Response

Recommendations

  • Clarify Accuracy Metrics:

- Resolve discrepancies in reported accuracies (88.8% vs. 93.5%) and consistently report the final performance across all sections.

answers: Thank you for your detailed comments. I have corrected them all to 88.8%.

  • Enhance CNN Details:

- Provide more information on the CNN architecture, loss functions, and hyperparameters and classification head used to balance the hardware-heavy focus.

answers: Thank you for your detailed comments. Please refer to lines 247-260 of Session 2, Table 5, lines 263-268, Table 6, lines 271-285, and figure 8 (all highlighted in gray). I have supplemented the CNN architecture, loss functions, hyperparameters, and classification head.

  • Address Visual Similarity:

- Experimentally validate proposed solutions (e.g., OCR, color histograms) to handle misclassifications of similar pills.

answers: Thank you for your detailed comments. Please refer to lines 169-171 of Session 2, Table 1, figure 6, lines 208-212, Table 2, lines 219-227, Table 3, lines 231-239, Table 4, lines 243-246. All highlighted in light blue.

Fix Figures and Equations:

- Correct redundant figures (e.g., Figure 6), remove duplicates (e.g., Figures 12, 13), and clarify every equation variable and their applications.

answers: I have removed the existing figures 12 and 13 and all the sentences about them. I have also replaced figure 6 with figures 4 and 6 and modified the sentences about them. Thank you.

The paper presents a promising system for automated pill classification, with strong integration of CNN and hardware components. However, inconsistencies in accuracy reporting, limited handling of similar pills, and grammatical issues detract from its quality. Addressing these weaknesses and implementing the recommended improvements would significantly enhance the manuscript’s impact and suitability for publication.

answers: The revisions have been reflected in the manuscript. Please refer to lines 491-497, 566-567 (all highlighted in purple) and table 11 of the Discussion.

Comments on the Quality of English Language

The paper’s English is generally clear but contains several issues that affect readability and professionalism. Below are key observations:

  • “precall” instead of “recall” in Equation (2) (Page 6).
  • answers: I have corrected it. Please refer to equation (2). I have also corrected the English grammar overall.
  • “CNN learning” is used instead of standard terms like “CNN training” or “deep learning” in multiple places (e.g., Page 1, Line 28).
  • answers: All corrected. Thanks for the feedback..
  • The paper would benefit from professional editing to improve flow and conciseness. For example, the abstract could be streamlined to avoid repeating “vibration amplitude” and “torque.”
  • answers: I briefly revised and organized the abstract.
  • Some sections, particularly the discussion (Pages 18–24), are verbose and could be condensed to focus on key findings and limitations.
  • answers: I have briefly revised and organized the discussion section as a whole.

Reviewer 2 Report

Comments and Suggestions for Authors

Thank you for your work. Two minor editing corrections: 

1. For figure 1 : Figure 1. This is a figure. Schemes follow the same formatting (a) imaging augmentation (b) imaging 78 classification (c) simulation results for the training accuracy.  I think you should  delete " This is a figure. follow the same formatting" and replace with "Image augmentation and classification" or similar. 

2. In figure 2 - the line from "merge with ..." block to "decision" - maybe to input in "decision" and should have an arrow from "merge .." to input in "decision"

Author Response

Comments 0 :

Thank you for your work. Two minor editing corrections:

Answer 0 : Thank you for your interest and evaluation in improving the quality of the manuscript. We have done our best to revise it based on your comments.

Comments 1 :

  1. For figure 1 :

Figure 1. This is a figure. Schemes follow the same formatting (a) imaging augmentation (b) imaging 78 classification (c) simulation results for the training accuracy.  I think you should  delete " This is a figure. follow the same formatting" and replace with "Image augmentation and classification" or similar. 

Answer 1 : Thank you, I have corrected it. Please refer to Figure 1.

Comments 2 :

 In figure 2 :

the line from "merge with ..." block to "decision" - maybe to input in "decision" and should have an arrow from "merge .." to input in "decision"

Answer 1 : Thanks for the detailed comment. I've corrected it for Figure 2(b).
